# Biased Deep Learning Methods in Detection of COVID-19 Using CT Images: A Challenge Mounted by Subject-Wise-Split ISFCT Dataset

**DOI:** 10.3390/jimaging9080159

**Published:** 2023-08-08

**Authors:** Shiva Parsarad, Narges Saeedizadeh, Ghazaleh Jamalipour Soufi, Shamim Shafieyoon, Farzaneh Hekmatnia, Andrew Parviz Zarei, Samira Soleimany, Amir Yousefi, Hengameh Nazari, Pegah Torabi, Abbas S. Milani, Seyed Ali Madani Tonekaboni, Hossein Rabbani, Ali Hekmatnia, Rahele Kafieh

**Affiliations:** 1Medical Image and Signal Processing Research Center, School of Advanced Technologies in Medicine, Isfahan University of Medical Sciences, Isfahan JM76+5M3, Iran; 2Law, Economics, and Data Science Group, Department of Humanities, Social and Political Science, ETH Zurich, 8092 Zurich, Switzerland; 3Institute for Intelligent Systems Research and Innovation, Deakin University, Melbourne, VIC 3125, Australia; 4Department of Radiology, School of Medicine, Isfahan University of Medical Sciences, Isfahan JM76+5M3, Iran; 5St. George’s Hospital, London SW17 0RE, UK; 6School of Engineering, University of British Columbia, Kelowna, BC V1V 1V7, Canada; 7Cyclica Inc., Toronto, ON M5J 1A7, Canada; 8Department of Engineering, Durham University, Durham DH1 3LE, UK

**Keywords:** COVID-19, deep learning, subject-wise data split, slice-wise data split, repeatability

## Abstract

Accurate detection of respiratory system damage including COVID-19 is considered one of the crucial applications of deep learning (DL) models using CT images. However, the main shortcoming of the published works has been unreliable reported accuracy and the lack of repeatability with new datasets, mainly due to slice-wise splits of the data, creating dependency between training and test sets due to shared data across the sets. We introduce a new dataset of CT images (ISFCT Dataset) with labels indicating the subject-wise split to train and test our DL algorithms in an unbiased manner. We also use this dataset to validate the real performance of the published works in a subject-wise data split. Another key feature provides more specific labels (eight characteristic lung features) rather than being limited to COVID-19 and healthy labels. We show that the reported high accuracy of the existing models on current slice-wise splits is not repeatable for subject-wise splits, and distribution differences between data splits are demonstrated using t-distribution stochastic neighbor embedding. We indicate that, by examining subject-wise data splitting, less complicated models show competitive results compared to the exiting complicated models, demonstrating that complex models do not necessarily generate accurate and repeatable results.

## 1. Introduction

The spread of corona virus disease in 2019 (COVID-19) led to a worldwide pandemic. This condition is caused by severe acute respiratory syndrome corona virus 2 (SARS-CoV-2). The gold standard diagnostic investigation for COVID-19 is the reverse-transcription-polymerase chain reaction (RT-PCR) test. Furthermore, chest computed tomography(CT) has also been accessible to clinicians for diagnosis [1]. The current literature has shown that artificial intelligence (AI) is a rapidly growing technological advancement, actively aiding the field of medical imaging in the fight against COVID-19. The traditional qualitative assessment of chest CT slices is very time-consuming and prone to error. CT data typically contain hundreds of slices that need to be read and assessed individually to reach an accurate diagnosis. Hence, AI-assisted deep learning (DL) techniques are preferred, and most of the recent publications have demonstrated promising outcomes on published datasets [2,3,4,5]. The main concern about the available models and the datasets is related to slice-wise splitting of the data, rather than focusing on subject-wise splits. In subject-wise data splits, all CT slices coming from one subject are allocated only to a test or training set, whereas in slice-wise split, this point is not considered [6]. Current studies of the application of DL in health-related data have shown that slice-wise splits overestimate the accuracy of the models and make them unreliable in clinical diagnostic applications [6,7]. In particular, multiple observations from each subject should be split only in the training set or the test set. Otherwise, complex DL algorithms are powerful enough to detect a confounding relationship between the subject and the diagnostic status. Accordingly, unrealistically high prediction accuracy may be reported, which should be avoided by focusing on subject-wise splitting of the dataset.

Considering the available CT datasets for the diagnosis of COVID-19, many well-known datasets are currently in the literature. Our analyses revealed that most publications have used slice-wise splitting, resulting in overestimated high reported accuracy (Table 1). Although in Table 1, we have presented some papers with subject-wise splits that have reported high performance among their outcomes, we indicate that the designed models in these papers typically use extremely selected and well-processed datasets that are not representative of real-world circumstances. We investigate this issue in more detail in the following sections.

To compare the proposed method with the published works and to show the disadvantage of reporting the results from slice-wise splits, we introduce a new publicly available CT slice dataset (ISFCT Dataset), accompanied by a subject-wise split, demographic data, and ground truth classification (https://zenodo.org/record/7997151, accessed on 2 June 2023). The ground truth is not only limited to COVID-19 and healthy classes, and it benefits from detailed information comprising eight characteristic lung features in the right or left lung for all CT slices from all subjects. Moreover, we applied a diverse range of previously published and new DL models to some of publicity existing datasets (elaborated in Table 1) and the new ISFCT dataset and evaluated the performance of the models using slice-wise and subject-wise strategies, generalization ability, and repeatability. We also evaluated the effect of decreasing or increasing the complexity of the models (for example, number of the layers) on the performance of the models. The utilized and newly designed DL models are freely available for further comparison by researchers (https://zenodo.org/record/7997151, accessed on 2 June 2023). It is expected that this dataset and models will improve reproducibility and make the associated research more discoverable with less need to duplicate efforts.

## 2. Materials and Methods

A comprehensive range of different algorithms in the literature was applied to the ISFCT dataset, and their performance was compared. Furthermore, two shallow models were developed to show how a simpler architecture could address our and other available datasets.

### 2.1. Dataset

The ISFCT Dataset was collected by Sepahan Radiology, Isfahan, Iran, and the study was approved by the Isfahan University of Medical Sciences Institutional Review Board (IRB) (IR.MUI.RESEARCH.REC.1399.003) and adheres to the tenets of the Declaration of Helsinki. Written informed consent was obtained from all participants.

A range of characteristic lung features is already known for COVID-19, including ground-glass opacity (F1),peripheral (F2), central (F3), peribronchovascular (F4), consolidation (F5), reversehalo (F6), crazypaving (F7), and atelectasis (F8) (eight subclasses) [25]. These radiological patterns are significant for determining the severity of COVID-19 [13]; however, in previous AI-based studies, such detailed features were all merged in the presence or absence of COVID-19. To address this issue in the ISFCT Dataset, eight characteristic lung features and their existence in the right (R) or left (L) lung were marked for each CT slice. Figure 1 and Table 2 demonstrate a summary and samples of the features. In binary classification, in which only health and COVID-19 are considered, the COVID-19 label is assigned according to presence of F1-R or F1-L, as is prevalent in other published works. According to previous studies, men appear to fare worse, and our findings showed a similar data distribution [26]. A previous study examining a mixture of patients suffering from COVID-19 and SARS observed that there are positive correlations of increasing age, male gender, and severity of disease with mortality, and our dataset agrees [27]. In accordance with our demographic data, the average age of patients infected by COVID-19 is 51 years old with a standard deviation of 14.61, and 57 percent of all patients are men.

The ISFCT Dataset contains data from 178 subjects diagnosed with COVID-19 (43,399 CT slices) and 156 healthy controls (39,767 CT slices). All CT slices have an equal size of 768 × 768 pixels, and each subject has (on average) 260 CT slices. The distribution of scans labeled with each of the eight characteristic lung features is presented in Figure 2. This dataset alsoincludes demographic details, including gender and age.

### 2.2. Data Preprocessing

Each individual is presented by multiple CT slices, and each CT slice may have one of the lung characteristic features in the left (L) or right (R) lung; the presence of each is considered presence of that feature in the CT slice. Many images contain more than one feature because of the characteristic lung changes during acute and chronic periods of disease [25]. The patterns observed on CT images of COVID-19 are dynamic and can be categorized into four stages. The earlystage spans from 0 to 4 days following the manifestation of the initial symptoms. In this stage, one can observe F1 unilaterally or bilaterally in the lower lobes. The second stage has a duration of 5–8 days and is known as the progressive stage, during which one can detect F1, F7 patterns, and F5. These features are distributed bilaterally in multiple lobes. The third stage, known as the peak, is from the ninth to the 13th day. During this period, dense F5 becomes more predominant. The fourth stage of the disease, absorption, becomes apparent after 14 days, around the time the infection becomes more controlled. At this stage, the F7 pattern and F5 are steadily absorbed. However, F1 remains. These radiological patterns of disease are significant for classification of severity of COVID-19 during assessment of the patients. As a result, some images are placed in different categories. In the supporting materials, a Microsoft Excel file is prepared, and for each patient, number 1 in each column (features) indicates the presence of any feature. The input CT slices are generally scaled to retain compatibility with the network structures. For most of the networks, the input size is fixed at 224 × 224 × 3. However, for some networks, such as EfficientNetB0 and Efficient CovidNet [33], the input dimensions are adjusted to the determined size by the published papers. To standardize the CT slices, the brightness of each image is transferred to the 0–1 interval. The different data split methods utilized in this paper are illustrated in Figure 3.

### 2.3. Algorithms Based on Transfer Learning

Most DL algorithms have numerous parameters, training of which requires a huge number of labeled data. Transfer learning is introduced to take advantage of such algorithms in applications with small numbers of data. For this purpose, a pre-trained DL algorithm with learned parameters is used, and only a small proportion of the parameters remained trainable with the new dataset (which is not essentially similar to the original data) [10,27]. We used famous pre-trained models (trained on the ImageNet dataset [34], including ResNet50V2, EfficientNetB0, EfficientNetB3,VGG16, EfficientNetB0, and EfficientNetB3 [35]). Additionally, a limited number of published algorithms trained with COVID-19 data (NASNet-based model [9], COVID-NET, and MobileNet) were used as a baseline for our transfer learning approaches.

For transfer learning, the kernels of the CNN layers were initialized with pre-trained ImageNet weights. For the EfficientNetB0, EfficientNetB3, ResNet50V2, VGG16, and VGG19 models, the weights of CNN levels of the models were frozen first, and all fully connected layers were removed and substituted with a new fully connected classifier. The architecture of this classifier was selected empirically. More details are provided in Table 3. For transfer learning on COVID-19-based networks, such as Efficient CovidNet, NASNet, and COVID-Net, the networks were pre-trained on ImageNet and then on provided COVID-19 datasets in their related papers.

### 2.4. Shallow Algorithms

To show the performance of shallow networks in addressing our and other datasets, we also developed M1- and M2-COVID networks. The M1-COVID network consists of three convolution operations. We adopted max-pooling after each convolution, and a batch-normalization layer was utilized. Rectified linear unit (ReLU) was considered as an activation function, and two dense layers were finally added. The M2-COVID network has similar structure to M1-COVID, with two additional convolutional layers, each followed by batchnormalization and max-pooling. Additionally, more dropouts between layers were added. In terms of hyperparameters, we employed Adam optimization and a batch size of 32 for both networks. The loss function was set to categorical cross-entropy.

### 2.5. Implementation Details

To train the models, the learning rate was initialized with a value of 0.0001. This value was determined by a grid search and then tuned using a cosine annealing learning rate scheduler [36,37] and a decay factor. In our dataset, the COVID-19 and non-COVID-19 case distributions were almost similar, but a weighting scheme was used to eliminate any bias in predictions, giving more weight to samples from smaller data groups. Softmax was used as an activation function in the last layer of all models. Additionally, the binary cross-entropy as a loss function and Adam as an optimizer showed better performance on examined models and were used to train the models.

For transfer learning models (based on networks pre-trained on ImageNet), the weights of all the layers of the base model were frozen, and the fully connected layers were removed and substituted by newly trained classifiers.

For networks pre-trained on COVID-19, two strategies were selected:For evaluating the performance of these models on our new dataset, the model was trained from scratch;For transfer learning, fully connected layers were substituted and retrained.

### 2.6. Performance Evaluation Metrics

Model accuracy and loss curves are presented for evaluation of the networks. For M1- and M2-COVID, confusion matrixes are also presented. The following metrics are al so provided to show the performance of the models:(1)$$Accuracy=\frac{TP+TN}{TP+FT+TN+FN}$$
(2)$$Specificity=\frac{TN}{TN+FP}$$
(3)$$Precision=\frac{TP}{TP+FP}$$
(4)$$Sensitivity/Recall=\frac{TP}{TP+FN}$$
(5)$$NPV=\frac{TN}{TN+FN}$$
where true positive (TP) refers to the correctly classified COVID-19 samples; false-positive (FP) is the number of non-COVID-19 samples classified as COVID; true-negative (TN) indicates the non-COVID slices correctly classified as non-COVID; and false-negative (FN) refers to the number of COVID-19 samples classified as non-COVID.

## 3. Results

### 3.1. Comparison of Algorithms in Slice-Wise and Subject-Wise Splits

In this section, various models are assessed in two distinct ways using our new ISFCT dataset:In slice-wise setting, the data are randomly divided, and CT slices from the same subject maybe present in both the training and test data;In subject-wise setting, the data are split such that the training and test splits contain slides from different subjects.

The evaluated algorithms include transfer-learning models and shallow algorithms. Table 4 shows the performance in more detail. Based on the results, COVID-Net, M1-COVID, and M2-COVID achieved the best results on the ISFCT dataset with a slice-wise split.

The COVID network based on NASNet also showed good performance. These results indicate that our simple models, M1-COVID and M2-CVOID, have good performance in comparison with investigated transfer learning-based models and multiple COVID-based networks.

It is clear from Table 4 that the performance of the models decreased drastically when the data were split subject-wise. Achieving high accuracy with a subject-wise split remains a challenging task.

To provide a better view regarding the high performance of our two shallow models, in Figure 4, ROC, learning rate, and precision-recall curves with different thresholds for M1-COVID and M2-COVID models in the slice-wise approach are shown.

To provide a visual description of different splitting methods, each CT slice is represented by two features (one point) using t-distributed stochastic neighbor embedding (t-SNE). CT slices pertaining to each subject are depicted in different colors, and 10 individually clusters are found for 10 randomly selected subjects in Figure 5, indicating common characteristics of slices from each subject. Figure 6 provides visual comparison of slice-wise and subject-wise splitting in CT slices. There is no doubt that discrimination of COVID (blue) from healthy (red) is a challenging task. However, as shown with slice-wise splitting (left panel—Figure 6), test images (circle mark) are selected from subjects (clusters), which are learned by the algorithm during the training stage (x mark). In particular, in slice-wise splitting, the leakage of information yields unreliably high accuracy, which is not achieved with a subject-wise split (right panel—Figure 6).

### 3.2. Average Accuracy of DL Models in Classification of Sub-Class Features

In this section, the performances of DL models in the classification of sub-class features are presented. We conducted a comprehensive examination of the provided eight features in the ISFCT dataset. Our main aim was to evaluate how effectively models could detect individual features, which are considered significant in COVID-19 detection but have received less attention in previous studies. In our evaluation, we assessed model performance for each feature (both right and left) using a multi-classifier approach. This approach determined the presence or absence of the examined feature in each sample. We present the results in Table 5. However, it is important to note that there were an insufficient number of samples available for features 6 and 7 to properly evaluate model performance. As a result, these two features are not included in the table. In general, our investigation revealed that the examined models did not exhibit remarkable performance in accurately identifying the different features present in the provided samples. Therefore, it is crucial for future research to focus on developing models that effectively detect these features. Additionally, further investigation should explore the impact of the presence of these features on detecting the given disease.

## 4. Discussion

COVID-19 detection papers frequently employ explainable methods, such as Grad-Cam [38], occlusion sensitivity maps [39], and GSInquire [40], to demonstrate whether the detection decision of their proposed model is based on relevant information. These techniques are used to ensure that models are not drawing conclusions based on irrelevant data, which might result in situations in which correct decisions are made for the wrong reasons. However, these tools only show the decision of a proposed model in a specific dataset, and they cannot be interpreted as tools to show generalizability of the models. If a model can find the cause of a disease such as COVID-19 by analyzing CT slices, it should perform well on similar data. We examined a number of publicly available COVID-19 datasets. In some datasets, such as [9], the subject to which each slide belongs to was not determined. We tested M1-COVID on the dataset provided in [9]. The results after only 15 epochs were very promising (accuracy 0.99, precision 0.99, and recall 0.8), offering the conclusion that the data are divided slice-wise and randomly. We also examined the COVIDxV9B dataset provided by [2]. The authors proposed COVID-NET to be applied on this dataset, which is divided into training and test tests in a subject-wise manner. We applied multiple models, such as VGG16, Resnet, DenseNet201, EfficientNetB0, and M2-COVID, on this dataset. All investigated models have shown more than 86% accuracy, demonstrating that the dataset consists of an easy set of images based on these findings. However, when we applied the provided model, COVID-NET, to our dataset, the model performed poorly. We conclude that the COVIDxV9B collection (which has been declared to be a subject wise split) does not reflect all CT imaging data collected in health centers, and almost all of the models have very good performance on it. Although these models provide many achievements and new understandings, they are not applicable in real-world scenarios. We came to the conclusion that models perform effectively for two reasons: they are trained on a biased dataset with the data partitioned slice-wise, or they are trained on datasets that are in an ideal shape and selected with bias. According to these findings, creating models capable of operating on heterogeneous datasets that are similar in content but vary in other features is crucial. Our publicly accessible data on eight sub-classes demonstrate enhanced performance with relatively small networks but not very high performance when subject-wise splitting is considered. We are providing free access to the ISFCT dataset. Moving forward, our work will focus on utilizing and analyzing the dataset using a subject-wise split. This approach will enable us to gather more reliable and accurate findings over time.

It is important to mention that, although the dataset is extensive, it does have its drawbacks. Some features, such as reverse halo and crazy paving, occur rarely, making it difficult to train networks with precision. Moreover, the dataset lacks proper scoring for each feature, creating ambiguity in determining the significance of each feature in classifying a patient as normal or infected. However, there is potential for creating a more comprehensive dataset in the future, enabling a more in-depth analysis of how each feature impacts disease diagnosis and assigning appropriate scores for these features.

## 5. Conclusions

In this paper, the ISFCT Dataset is introduced to challenge the previously published biased works, which were designed without considering subject-wise splitting. Furthermore, each CT scan in this new dataset has more than only two labels (COVID-19 and healthy). In particular, F1 toF8 are introduced as features in the lung, and their presence in the right or left lungs is labeled.

Our primary focus in this work is to classify each subject-wise split CT scan to a healthy or COVID-19 set (based on the presence of F1-R or F1-L). We examine multiple models on our dataset using two methods of data splitting, slice-wise and subject-wise. With a slice-wise approach, data are randomly split into training and test sets. As a result, CT slices from the same individual are possibly present in both the training and test sets, which is a kind of information leakage and can lead to bias. The examined models include transfer learning using the most successful models and some state-of-the-art COVID-19 detection networks, as well as two shallow models. We show that shallow models perform acceptably well in a slice-wise split, with results comparable to state-of-the-art COVID-19 detection models. However, when the data are divided subject wise (such that the training and test sets comprise CT slices from different subjects), the performance of all developed models decreases drastically. We therefore conclude that the high performance of many published works is only due to slice-wise splits and leakage of the information, rather than the power of the designed networks, which perform similar to our proposed shallow models (M2). Generally, the bias problem in DL models due to incorrect data distribution is difficult to resolve with transfer learning or augmentation [41]. The comparably high performance of the proposed shallow network (M2) demonstrates that a deeper model does not always imply a more unbiased and accurate model. Furthermore, the model’s generalization does not necessarily improve by training models on out-of-field datasets. This finding shows that evaluated models that have shown promising results in previous papers were only customized to perform well with specific datasets and not in most cases, in which the data are not split in a subject-wise manner.

## Figures and Tables

**Figure 1 jimaging-09-00159-f001:**
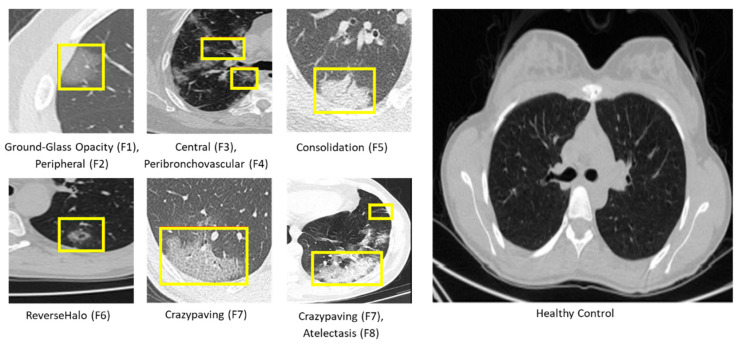
Samples of eight characteristic lung features and one healthy case.

**Figure 2 jimaging-09-00159-f002:**
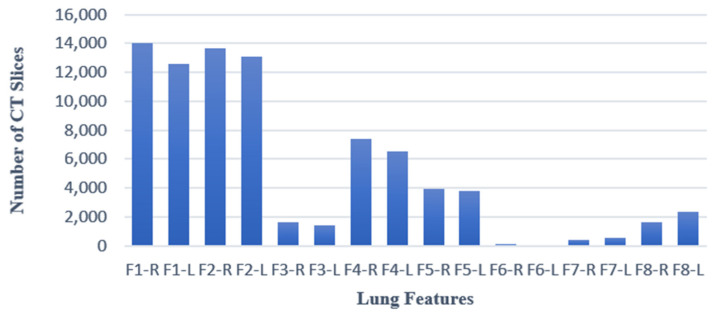
Distribution of CT slices in each subclasses (eight characteristic lung features).

**Figure 3 jimaging-09-00159-f003:**
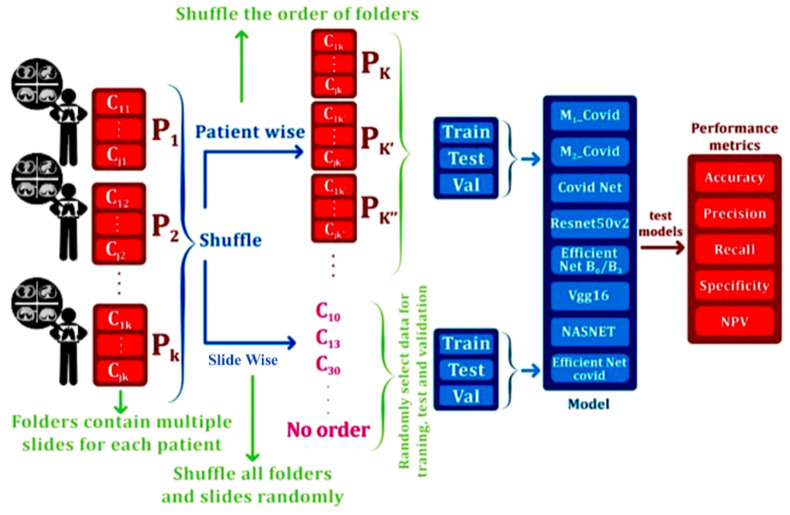
The proposed process for data selection and evaluation.

**Figure 4 jimaging-09-00159-f004:**
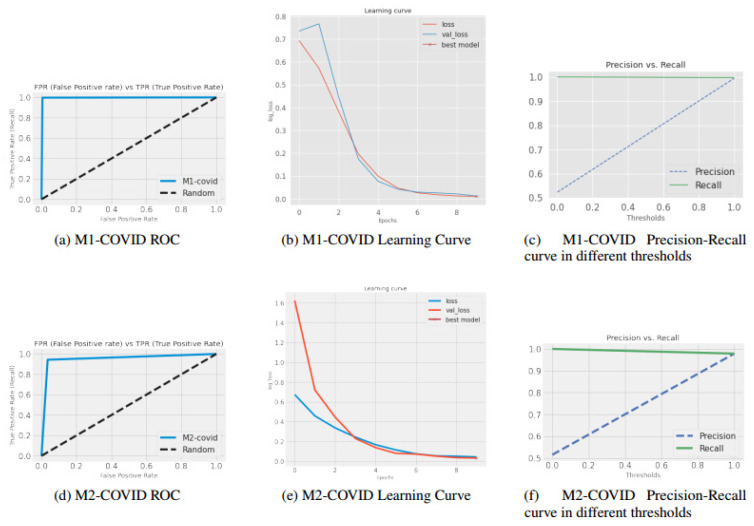
(**a**,**d**) ROC, (**b**,**e**) training, and validation learning curves, and (**c**,**f**) precision/recall curves for M1-COVID and M2-COVID using various thresholds.

**Figure 5 jimaging-09-00159-f005:**
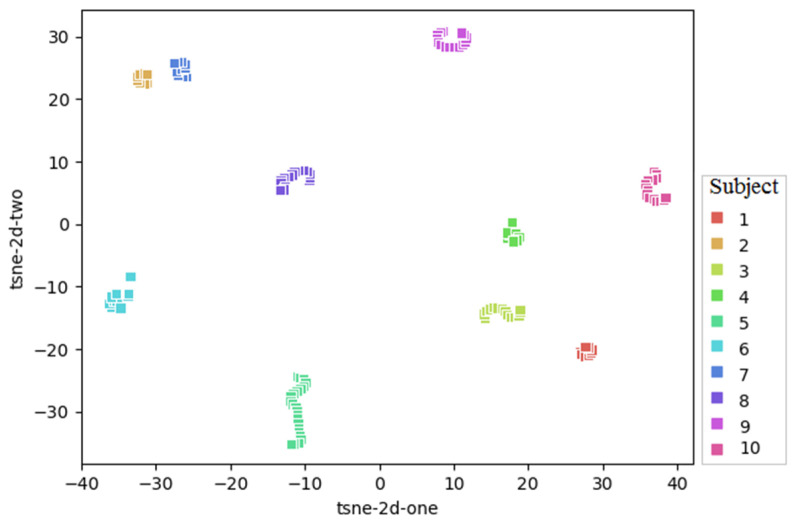
Visual presentation of common characteristics between the CT slices of each subject. Each CT slice is represented by two features (one point) using t-distributed stochastic neighbor embedding (t-SNE). CT slices pertaining to each subject are depicted in different colors, and 10 individual clusters are found for 10 randomly selected subjects.

**Figure 6 jimaging-09-00159-f006:**
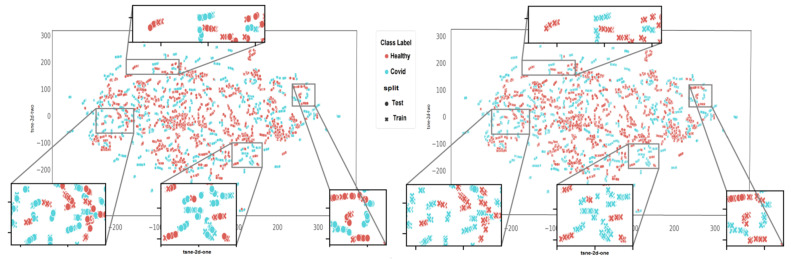
Visual comparison of the slice-wise and subject-wise splitting in CT slices. The goal is discrimination of COVID (blue) from healthy (red) data. In slice-wise splitting ((**left**) panel), test images (circle mark) are selected from the subjects (clusters), which are learned by the algorithm during the training stage (x mark). In particular, in slice-wise splitting, the leakage of information yields unreliably high accuracy, which is not achieved with a subject-wise split ((**right**) panel).

**Table 1 jimaging-09-00159-t001:** Summary of the published works on usage of DL in diagnosis of COVID-19 with CT slices (slice-wise = IW, subject-wise = SW, C = COVID-19 and NC = non-COVID-19, N = normal, ND = not determined).

Published DL Work	Dataset	Slice-Wise or Subject-Wise	Number of CT Slices	Number of Subjects	DL Models	Reported Accuracy
Yang et al. [8]	COVID-CT dataset [8]	IW	C: 349 NC: 463	C: 216 NC: VD	Transfer-Learning: DenseNet-169, ResNet-50, and contrastive self-supervised transfer learning	F1-Score = 0.90 AUC = 0.98 Accuracy = 0.89
Ghaderzadeh et al. [9]	Ghaderzadeh et al. [9]	IW	C: 7644 NC: 2509	C: 190 NC: 59	Transfer-Learning: NASNet	Sensitivity = 0.999 Specificity = 0.986 Accuracy = 0.996
Zhao et. al [10]	COVIDx CT 2A [11]	IW	C, SARS-CoV-2 and CP: 194,922 N: ND	C, SARS-CoV-2, and CP: 3475 N: ND	BigTransfer (BIT) [12]	Accuracy = 0.992 Sensitivity = 0.987 Specificity = 0.995 NPV= 0.996 PPV=0.985
Zhang et al. [13]	CC-CCll; Zhang et al. [13]	IW	C: 156,070 viral pneumonia: 159,700 N: 95,459	C: 839 viral pneumonia: 874 N: 758	A 3D classification networks	Accuracy = 0.924 Sensitivity = 0.943 Specificity = 0.911
Kassania et al. [14]	Cohen et al. [15], and KaggleRSNAPneumonia Detection dataset [16]	IW	C: 20 N: 20	C: ND N: ND	Examination of transfer learning with a range of methods, including: DenseNet121, Xception, InceptionV3, DenseNet201, InceptionResNetV2, VGG16, VGG19, NASNETLarge, NASNetMobile, ResNet50v2, ResNet101V2, and ResNet152V2	Accuracy = 0.99 Precision = 0.99 Recall = 0.99 F1-Score= 0.99
Jaiswal et al. [17]	COVID CT slices [8]	IW	C: 746 N: ND	C: ND Normal: ND	COVIDPEN: Pruned EfficientNet-B0	Accuracy = 0.85 AUC = 0.84 F1-Score = 0.86
Kogilavani et al. [18]	Kaggle	IW	C: 1958 N: 1915	C: ND N: ND	VGG16, DenseNet, MobileNet, Xception, EfficientNet, and NASNet	Accuracy = 0.97 F1-Score = 0.98 Precision = 0.99 Recall = 1
Chouat et al. [19]	GitHub repository [8], and Kaggle [20,21]	IW	C: 408 N: 325	C: ND N: ND	VGGNet-19, ResNet50, Xception, and InceptionV3	Accuracy = 0.905 F1-Score = 0.905 Precision = 0.915 Recall= 0.903
Zouch et al. [22]	Database of CT slices provided in GitHub [23]	IW	C: 349 NC: 408	C: ND NC: ND	VGG19 and ResNet50	Accuracy = 0.98 F1-Score = 100 Precision = 0.993 Recall = 100
Ortiz et al. [24]	CC-CCll; Zhang et al. [13]	SW	C: 156,070 viral pneumonia: 159,700 N: 95,459	C: 839 viral pneumonia: 874 N: 758	Inception ResNetV2	Accuracy = 0.95 AUC = 0.96 Sensitivity = 0.94
Wang et al. [2]	COVIDx [2]	SW	C: 358 CXR N: not determined	C: 266 NC: 5538 no pneumonia: 8066	COVID-Net	Accuracy = 0.933 Sensitivity = 0.91 PPV = 0.989

**Table 2 jimaging-09-00159-t002:** The explanation of eight characteristic lung features.

Feature	Explanation	Symbol
Ground-Glass Opacity	The hazy gray indicates increased density inside the lungs [25].	F1
Peripheral	The feature is situated on the edge or periphery of the lung.	F2
Central	The feature is located in the middle of the lung.	F3
Peribronchovascular	Thickening of the interstitial or bronchial wall [28].	F4
Consolidation	The alveolar air spaces are filled with fluid, cells, tissue, or other material [29].	F5
Reverse Halo	Central ground-glass opacity surrounded by denser consolidation of a crescent shape or a complete ring of at least 2 mm in thickness [30].	F6
Crazy Paving	Scattered diffuse ground-glass attenuation with superimposed interlobular septal thickening and intralobular lines [31].	F7
Atelectasis	Complete or partial collapse of the entire lung or area(lobe) of the lung [32].	F8

**Table 3 jimaging-09-00159-t003:** Architecture of different transfer learning models (FC: demonstrating how fully connected layers have been developed).

Classification Models	Input Slice Size	FC	Initial Weights
ResNet50v2	(224 × 24 × 3)	AveragePooling2D Flatten layer Dense 256 Drop out 0.5 Dense 2	ImageNet
VGG16	(224 × 224 × 3)	MaxPooling2D Flatten Dense256 Dense 2	ImageNet
EfficientNetB0	(224 × 224 × 3)	MaxPooling2D Flatten Dense256 Dense 2	ImageNet
EfficientNetB3	(300 × 300 × 3)	MaxPooling2D Flatten Dense256 Dense 2	ImageNet
MobileNet	(224 × 224 × 3)	AveragePooling2D Flatten layer Dense 256 Dropout 0.5 Dense 2	ImageNet
COVIDnetwork based on NASNET	(224 × 224 × 3)	No change	ImageNet and Ghaderzadeh et al. [9] COVID-19 dataset
COVID-Net	(224 × 224 × 3)	Dense 2	COVIDx [2]

**Table 4 jimaging-09-00159-t004:** Performance of the examined models based on different measures.

Models		Slice-Wise					Subject-Wise				
		ACC	Precision	Recall	Specificity	NPV	ACC	Precision	Recall	Specificity	NPV
Transfer Learning Models	ResNet50V2	0.84	0.64	0.46	0.76	0.60	0.59	0.35	0.11	0.84	0.55
	EfficientNetB0	0.52	0.50	1	0	0	0.44	0	0	1	0.56
	EfficientNetB3	0.50	0	0	1	0.50	0.48	0	0.4	0.6	0
	VGG16	0.84	0.73	0.38	0.87	0.60	0.62	0.69	0.37	0.79	0.55
	COVID network based on NASNET	0.94	0.93	0.95	0.92	0.95	0.66	0.66	0.67	0.67	0.66
	MobileNet	0.52	0.52	0.56	0.47	0.50	0.49	0.47	0.40	0.57	0.50
	COVID-NET	0.99	1	0.99	1	0.99	0.60	0.73	0.54	0.7	0.50
Shallow Models	M1-COVID	0.99	0.99	0.99	0.99	1	0.58	0.58	0.40	0.56	0.58
	M2-COVID	0.99	1	0.98	1	0.98	0.61	0.67	0.40	0.78	0.53

**Table 5 jimaging-09-00159-t005:** The average accuracy of different models in sub-classes.

Models		F1	F2	F3	F4	F5	F8
Transfer Learning Models	ResNet50v2	0.75	0.57	0.50	0.47	0.57	0.51
	EfficientNetB0	0.73	0.50	0.53	0.48	0.51	0.37
	EfficientNetB3	0.74	0.40	0.69	0.40	0.48	0.40
	VGG16	0.74	0.45	0.65	0.46	0.64	0.64
	COVID network based on NASNET	0.80	0.53	0.76	0.56	0.72	0.66
	MobileNet	0.67	0.53	0.60	0.53	0.51	0.68
	COVIDNET	0.75	0.40	0.79	0.61	0.65	0.60
Shallow Models	M1-COVID	0.73	0.50	0.77	0.71	0.74	0.40
	M2-COVID	0.50	0.50	0.62	0.58	0.59	0.52

## Data Availability

Our dataset containing COVID-19 and normal cases, as well as the demographic details, is available at https://zenodo.org/record/7997151 (accessed on 2 June 2023), and the utilized and newly designed DL models are freely available for further comparison by researchers (https://zenodo.org/record/7997151, accessed on 2 June 2023).

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
