# Peer review of "Biased Deep Learning Methods in Detection of COVID-19 Using CT Images: A Challenge Mounted by Subject-Wise-Split ISFCT Dataset"

_2313-433X, 2023, doi:10.3390/jimaging9080159_

Round 1
Reviewer 1 Report
The topic seems to be relevant to a current problem for AI concept (relevant datasets).
The paper is relevant, but it can be improved.
Comparing to other papers or topics, paper's contribution can be improved. Other datasets are available on the Internet and are well structured as well.
As a specific improvement I suggest using this dataset with a new AI architecture developed by the authors.
Conclusions present a good comaprison with other papers.
I recommend for publishing in current form.
Quality of English is good enough in current form.
I recommend a more scinetific lexicon in future papers.
Author Response
The response to reviewers is attached.

Reviewer 2 Report
Thanks for their contribution and conclusion about COVID-19 research and background research accuracy. Their objective was to detect COVID in a subject-wise CT setting, which is very nice. Actually, that gives a clear idea about model efficiency. Before going to the next step, the following concerns need to be addressed:
1. Line 155, incomplete sentence
2. Table 4, please check the vgg-precision value of "073."
3. What conclusions can be drawn from Figures 4-5?
4. Line 250 should be 3. 2. In this sub-section, why did they implement 4-class classification using only F1-L, F1-R, F3-L, and F3-R? Why not other classes? How can they compare with full-dataset performances? Need more details about this section.
5. In Table-2, FC layers and Input shape should be identical to compare among them.
6. I'm suggesting that we separate the discussion and conclusion. And highlights the changes in a point base with the discussion. The future direction should be concluded.
7. In Line 42, I also suggest two articles as DL applications in the medical imaging field, https://doi.org/10.1145/3378936.3378968 and https://ieeexplore.ieee.org/document/9121222
Minor changes required
Author Response
The response to reviewers is attached.
Reviewer 3 Report
The authors used famous pre-trained models ResNet50V2, EfficientNetB0, EfficientNetB3, and VGG16. EfficientNetB0 and EfficientNetB3, and some new as M1- and M2-COVID networks were built.
For all CNNs the accuracy of classification was performed; for famous pre-trained CNN models the metrics extracted from the confusion matrix are lower than M1- and M2-COVID CNNs.
The results are compared with others from the scientific literature.
The proposed study is based on 40 new references.
Appendix A should be added to the results.
The quality of the figures is bad; these should be improved.
Please add the conclusion as a separate section.
Author Response
The response to reviewers is attached.

Round 2
Reviewer 2 Report
I'm not happy with their response and revised version. In a few queries, they have avoided the main point. The author should attach my response separately and more accurately. I found a few mismatches between the response and the revised version. I'm suggesting that the author revise the manuscript thoroughly.
1. Lines 244–247 should be Fig. 6. They say Fig. 5.
2. For Fig. 4, authors must describe sub-images. e.g., 4(a) represents ROC, 4(b) models train and val losses, and so on.
3. On line 305, where are the limitations and implications of the
findings? Need to clear
4. Conclusion should be the last section of this manuscript.
